# Element Levels and Predictors of Exposure in the Hair of Ethiopian Children

**DOI:** 10.3390/ijerph17228652

**Published:** 2020-11-21

**Authors:** Maria Luisa Astolfi, Georgios Pietris, Corrado Mazzei, Elisabetta Marconi, Silvia Canepari

**Affiliations:** 1Department of Chemistry, Sapienza University, Piazzale Aldo Moro 5, I-00185 Rome, Italy; silvia.canepari@uniroma1.it; 2Department of General Surgery, Thoracic Diseases General Hospital Sotiria of Athens, Mesogion 152, 115 27 Athens, Greece; gpietris@gmail.com; 3Canon Toshiba Medical Systems s.r.l., Via Canton 115, I-00144 Rome, Italy; corradomazzei1964@gmail.com; 4Department of Public Health and Infectious Diseases, Sapienza University, Piazzale Aldo Moro 5, I-00185 Rome, Italy; elisabetta.marconi@uniroma1.it

**Keywords:** biomonitoring, human hair, children, trace elements

## Abstract

Children’s development and health may be affected by toxic heavy metal exposure or suboptimal essential element intake. This study aimed to provide updated information regarding the concentrations of 41 elements in children’s hair (aged under 18) living in a rural area of the Benishangul-Gumuz region, Ethiopia. The highest average levels (as a geometric mean) for toxic heavy metals were obtained for Al (1 mg kg^−1^), Pb (3.1 mg kg^−1^), and Ni (1.2 mg kg^−1^), while the lowest concentrations among the essential elements were found for Co (0.32 mg kg^−1^), Mo (0.07 mg kg^−1^), Se (0.19 mg kg^−1^), and V (0.8 mg kg^−1^). Hair analysis was combined with a survey to evaluate relationships and variations among subgroups and potential metal exposure predictors. Females showed significantly higher concentrations for most hair elements, excluding Zn, than males, and the 6–11 years age group reported the highest levels for Be, Ce, Co, Fe, La, Li, Mo, and Na. The main predictors of exposure to toxic elements were fish consumption for Hg and drinking water for Ba, Be, Cs, Li, Ni, Tl, and U. The data from this study can be used to develop prevention strategies for children’s health and protection in developing countries.

## 1. Introduction

Elements present in the environment can come from natural (such as volcanic activity and forest fires) or anthropogenic (such as industrial, agricultural, and domestic) sources [1,2,3,4,5]. Factors that influence human exposure and, consequently, the presence of possible toxic effects are mainly dietary habits (i.e., fish, meat, cereals, vegetables, and water consumption) and outdoor and indoor air quality [6,7,8,9,10,11,12,13]. In particular, toxic heavy metals have no dietary value in humans [12,13]. More specifically, As, Hg, Pb, and Cd have been classified as the most dangerous elements affecting health [14]. As and Cd are classified as carcinogens (Group I), and Pb is ranked as a probable carcinogen (Group IIA) in humans by the International Agency for Research in Cancer (IARC) [15]. Instead, Hg, in organic [as methylmercury (MeHg)] or inorganic form, is classified by the IARC as a probable carcinogen in humans (Group IIB) or not classifiable as carcinogenic (Group III), respectively [15]. Instead, essential elements (such as Ca, Co, Cu, Fe, K, Mg, Mn, Mo, Na, P, Se, V, and Zn) are fundamental for the growth and health of children and perform several important functions in the human body, including bone formation, regulation of body fluids, and participation in the vital processes of cells [16,17]. However, these elements can also become harmful to human health if taken in excess from food or abnormal exposure [7,18,19]. Children exposed to environmental contaminants are more susceptible than adults, mainly due to their immature organs and differences in exposure [20,21]. Nutritional aspects (children eat more food and drink more water per unit of body weight than adults), rapid growth, active time spent outdoors, and specific behaviors (such as the tendency to place items in their mouths and contact with the ground) increase the risk of exposure in children [22]. In this regard, it is important to underline that exposure to low doses of trace metals can cause neurobehavioral and cognitive changes in children, even below concentrations considered safe for most people [23]. Human biomonitoring studies allow us to evaluate human exposure to elements through the measurement of chemicals in body fluids and tissues, such as blood [24,25,26,27], plasma [28,29], serum [30,31], breast milk [32,33], urine [34,35], saliva [36], lung fluids [37,38], nails [27,39], and also hair [40,41,42,43,44]. The latter has several advantages over other biological matrices. Among these, the first concerns the hair’s ability to bind various elements due to thiolic groups’ presence in their structure [45]. Elements can be accumulated in the hair at higher concentrations and for longer than other biomarker media such as urine and blood [46,47,48,49,50,51]. Hair grows at a rate of ~1 cm/month; therefore, the chronological exposure to elements can be traced from the segmental hair analysis over a defined period (at least 180 days) depending on the selected hair length [52]. The hair’s ability to sequentially accumulate chemicals in its inner structure, together with the opportunity to conduct retrospective analyses, means that hair analysis can be used for screening and confirmation purposes in various application contexts, such as forensic and clinical [53]. Another advantage when using hair compared to other biological matrices is that the sampling is painless and non-invasive and does not require experienced personnel [46,47]. Finally, the hair samples can be transported and stored at room temperature, and small specimen sizes are required for analysis [41,46,47]. Despite these advantages, hair analysis presents a few limitations, such as a lack of reference concentration ranges or difficulties in interpreting the results due to the presence of potential confounding factors like gender, age, hair color, dietary habits, living site, and lifestyle [54,55]. Hair results are a useful screening tool for exposure assessment, investigation of the development and state of nutrition, and possible pathological processes [55,56,57]. Furthermore, total Hg in hair can be used to assess exposure to MeHg because more than 80% of the total Hg analyzed in hair is present in organic form as MeHg [47,58,59].

This study aimed to assess the levels of essential and toxic heavy metals in the hair of children living in the Benishangul-Gumuz region, a Developing Regional State of North-Western Ethiopia. The influence of several factors in the variability of element concentrations in children’s hair (age, sex, body mass index, passive smoking, and eating habits) was also studied. To our knowledge, this is the first study in which essential and toxic elements are determined in the hair of Ethiopian children. We compared our results with other biomonitoring studies on children in the literature and studied the main predictors of exposure.

## 2. Materials and Methods

### 2.1. Design and Study Population

A cross-sectional study was conducted between November and December 2019 in Bameza, a rural area of the Benishangul-Gumuz region along the Blue Nile river in north-western Ethiopia (Africa) (Figure 1). Bameza is approximately 150 km from Benishangul-Gumuz’s capital, Asosa, and 500 km from Ethiopia’s capital, Addis Ababa. The study area landscape is undulating and covered by a thick savannah and forest (Appendix A). Due to a lack of communications infrastructure and transportation, the possibilities to travel within the Benishangul-Gumuz region are often scarce [60], and access to food and health services is inadequate [61]. Moreover, inadequate access to safe drinking water (mainly for microbiological quality), poor sanitation, low educational level, widespread poverty, and the highest risk of under-five mortality characterize the Benishangul-Gumuz region [62,63,64,65,66,67]. Improperly protected water collection and storage containers contribute to low drinking water quality [62].

The hair samples (*n* = 81) were taken from children aged under 18 (40 boys, 49.4%; 41 girls, 50.6%) whose hair had not been colored or treated. The study population’s characteristics were collected using individual anonymous questionnaires with specific information on age, gender, weight, height, passive smoking, and dietary habits (water, meat, fish, vegetables, fruit, legumes, cereals consumption). The variables studied are shown in Table 1. Food consumption frequency by food groups was converted into quantitative intakes (g month^−1^ or mL month^−1^), according to other authors [57,58,68]. Monthly intake of individual foods was estimated according to the following formula:FI = W or V × IFr(1)
where FI is the food intake (g month^−1^ or mL month^−1^), W = weight (g) or V = volume (mL) of the portion size, and IFr = intake frequency (number of portions) per month.

This research was a non-interventional/observational study based on the definitions of the European Directive 2001/20/EC for which the approval of an Ethics Committee was not requested [69]; it was conducted according to the Helsinki Declaration (1964) and its later amendments and followed the International Code of Ethics for Occupational Health Professionals [70]. Collected information was used on aggregate health data of the children, with no possibility of individual identification. Before collecting the questionnaire information and the children’s hair samples, all parents were informed about the study’s aim and gave their consent. Children’s hair sampling was performed at least with the presence of one of the parents.

### 2.2. Sample Collection

To avoid external contamination, hair samples (~0.05 g; ~1 cm long) were collected from the nape using stainless steel scissors and disposable vinyl gloves, as previously described [71,72]. Then, hair samples were stored in polyethylene bags until analysis at room temperature. 

We also measured elements in the following drinking water: the Blue Nile river and the Blue Nile river’s treated water. The water samples were collected at the same time as the hair samples in plastic urine collection cups, which were stored at −20 °C until analysis.

### 2.3. Chemical Analysis

#### 2.3.1. Hair Samples

Forty-one elements were analyzed, including essential of Ca, Co, Cr, Cu, Fe, K, Mg, Mn, Mo, Na, P, Se, V, and Zn, and potentially toxic of Al, As, B, Ba, Be, Cd, Cs, Hg, Li, Ni, Pb, Sb, Sn, Tl, and U (see Table 1). Chemical analysis was performed in the Chemistry Laboratory of Sapienza University of Rome (Italy). Except for Hg, element concentrations in the hair were evaluated using an inductively coupled plasma mass spectrometer (ICP-MS; 820-MS Bruker, Bremen, Germany) equipped with a collision-reaction interface (CRI) and glass nebulizer (0.4 mL min^−1^). The data were collected according to a previously reported method [71]. Mercury was analyzed using a cold vapor generation atomic fluorescence spectrometer (CV-AFS; AFS 8220 Titan, FullTech Instruments, Rome, Italy), as described previously [72,73]. The ICP-MS analysis mode, measured isotopes, and internal standards are shown in Appendix A, while the preparation of both calibration and internal standards is described in Appendix A.

Hair samples were digested using an analytical procedure described previously [71,73]. Briefly, 0.02 g hair samples were transferred into polypropylene tubes, mixed with 0.5 mL 67% HNO_3_ (super-pure, Carlo Erba Reagents, Milan, Italy) and 0.25 mL 30% H_2_O_2_ (super-pure, Merck KGaA, Darmstadt, Germany), and heated in a water bath system (WB12, Argo Lab Modena, Italy) for 20 min at 95 °C. The digest was left to cool down, and the contents of the tubes were diluted to 10 mL with deionized water (resistivity, ≤18.3 MΩ cm), filtered (0.45 μm pore size, GVS Filter Technology, Indianapolis, IN, USA), and analyzed by ICP-MS and, after further dilution in the ratio 1:1 with 6% HCl (assay >36%; Promochem, LGC Standards GmbH, Wesel, Germany), by CV-AFS. Each sample was analyzed in duplicate. Blanks were treated as samples for the subtraction of the background signal from the reagents.

The detection limits (DLs) established for each element ranged between 0.00002 (U) and 30 (K) mg kg^−1^. Inter-day precision [relative standard deviation (RSD), %] calculated through different days was below 13%, trueness bias ranged between −4 and 9%, and recoveries ranged between 90 and 110% for all the elements except for Cr (65%), and Fe (79%) [71,73].

#### 2.3.2. Water Samples

Drinking water sampling and analyses were carried out following the Italian reference analytical methods [74,75]. Briefly, a volume of 250 mL (three replicates) of both treated and untreated water of the Blue Nile river was collected using a sterile polyethylene container with a screw cap. Each sample had an indelible identification label to identify it uniquely. Sampling was carried out where households obtained their drinking water. In particular, if there were more possibilities of access to the same water source (different access points to the Blue Nile river or different taps of the same treated water), a single sampling point was considered (Appendix A). After sampling, all water samples were transported to the laboratory using thermal bags and then filtered, acidified to 2% (*v/v*) HNO_3_ (pH < 2), and stored at a temperature in the range 1–10 °C up to instrumental analysis. All water analyses were completed within one week after fieldwork. The water samples were analyzed diluted in a 1:10 and 1:100 ratio with 2% (*v/v*) HNO_3_ and without dilution. Blanks and calibration standards were also made in 2% (*v/v*) HNO_3_. Internal standards (45Sc, 89Y, 103Rh, 115In, 232Th) were added to all samples, blanks, and calibration standards for ICP-MS analysis. To verify the accuracy, certified reference material (SRM 1643e trace elements in water; National Institute of Standards and Technology, NIST; Gaithersburg) was analyzed. The percent relative standard deviation for the repeatability did not exceed the 10% limit, and trueness bias percentages in the range −5 to 10% for all the studied elements were found.

### 2.4. Statistical Analysis

The statistical analysis was performed using SPSS version 25.0 for Windows (SPSS Inc., Chicago, IL, USA). Levels below DL were replaced by DL/2 [76]. Elements with a concentration <DL in more than 20% of the samples were excluded from the statistical calculation. 

The most important descriptors such as arithmetic (AM) and geometric mean (GM), minimum and maximum levels, and the 25th, 50th, 75th, and 95th percentile were calculated to perform the descriptive statistical analysis. 

The Shapiro–Wilk test and the Levene test were used to evaluate the normality and equal variances hypotheses, respectively [77]. All element concentrations were not normally distributed, so non-parametric methods such as the Mann–Whitney test and the Kruskal–Wallis test were used to compare element levels according to variable categories [78]. 

The relationship between the hair elements’ concentrations and the factors considered was studied using the Spearman correlation test. For a value of *p* ≤ 0.05, a statistically significant relationship was considered.

A multiple linear regression model (backward method) was used to evaluate the influence of the study factors (gender, age, BMI, passive smoking, and water, meat, fish, fruit and vegetables, and cereals consumption) on the level of elements in children’s hair. In each model, the natural log-element concentration was included as a dependent variable. The *p* value used as a criterion for the entry and stay of the variables was 0.05 and 0.1, respectively. The 95% confidence intervals were calculated for the model coefficients to evaluate the sample’s statistical estimates [79,80]. The effect of potential outliers was checked using residual graphs.

## 3. Results and Discussion

### 3.1. Element Levels in Children’s Hair

Descriptive statistics of element levels are given in Table 2. 

Elements with concentrations lower than the DL in more than 20% of the hair samples were excluded from the statistical analysis [Al (40.7%), As (35.8%), B (79.0%), Bi (44.4%), Nb (65.4%), Si (23.5%), Te (66.7%), and W (86.4%)]. The highest concentrations were found for the essential elements with Ca, Fe, K, Mg, and Na having a GM higher than 100 mg kg^−1^, followed by P and Zn, which exhibited values of 98 and 86 mg kg^−1^, respectively. Average concentrations (as GM) above 1 mg kg^−1^ were found, in order of abundance, for Si > Mn > Sr > Ti > Cu > Ba > Pb > Rb > B, and Ni, while element concentrations ≤ 1 mg kg^−1^ were detected for all other elements with decreasing contents in the following order: Ce > V > Cr > La > Zr > Co > Sn > Li > Se > Cd > Ga > Sb > Mo > Hg > As > U > Cs > Be > Tl > Te > Bi > Nb > W. 

The possible exposure sources to the toxic elements analyzed may be several, such as contaminated food, water, and soil pollution, especially by the abuse or misuse of chemicals in agriculture, indoor and outdoor air pollution from inefficient solid fuel combustion (wood, animal dung, charcoal, crop wastes, and coal) and burning of waste, residues of manufacturing industrial products, mining, and oil refining [81,82,83]. As, Cd, Hg, and Pb are known to be of health concern in Africa, including in Ethiopia [81]. Figure 2, Figure 3, Figure 4 and Figure 5 show a comparison of the mean levels found in this study for the priority and most toxic trace elements (As, Cd, Hg, and Pb, respectively) with those of other biomonitoring studies in children.

Regarding As (Figure 2), the average level (GM: 0.04 mg kg^−1^) was similar or lower than those reported in other populations of Bangladesh [median (5–95th percentile): 0.53 (0.14–2.9) mg kg^−1^] [84], Greece (GM: 0.020–0.036 mg kg^−1^) [85], Russia in Moscow [median (25–75th percentile): 0.021] [86], and in both exposed and unexposed populations (median: 0.02 and 0.03 mg kg^−1^, respectively) [87]. Water, food, air, and soil can be possible exposure sources to As from both natural geochemical processes (erosion) or anthropogenic pollution (arsenical insecticides, improper waste and sewage disposal, combustion of fossil fuels) [81]. For Cd (Figure 3), higher levels were obtained (GM: 0.1 mg kg^−1^) compared with data reported in children from all the reported studies, excluding those from Russia, in both exposed and unexposed populations (median: 0.11–0.12 mg kg^−1^) [87]. Exposure to Hg can occur mainly by the ingestion of contaminated fish (organic Hg, such as methylmercury) and small-scale artisanal gold mining (elemental Hg) [81]. Furthermore, gold extraction is routinely carried out near water sources, contaminating the environment and drinking water. The overall Hg concentrations ranged from 0.012 to 0.483 mg kg^−1^ with a GM of 0.056 mg kg^−1^, which were the lowest compared with all data reported by other authors (Figure 4) [22,57,85,86,87,88,89,90,91,92,93]. All the Hg hair data were also below the health-based values proposed (0.58, 1.0, and 2.3 mg kg^−1^) [59]. However, the Pb levels (GM: 3.1 mg kg^−1^; Figure 5) were the highest compared with all data reported in other studies [22,55,84,85,86,87,94]. Hair can be used as a suitable biomarker of Pb exposure [95,96]. The reduction or removal of Pb from gasoline has produced a significant decline in pediatric morbidity. Llorente Ballesteros et al. [94] highlighted that the Pb content reduction in petrol was seen in environmental levels and, therefore, in human hair. However, children and adults continue to be exposed to Pb in Africa [81]. Here, most of the Pb in the environment comes from human activities such as waste combustion, mining, indiscriminate dumping, and even the use of Pb-based paints [81].

Some essential element average levels (Ca, Co, Cu, Mg, Mo, P, Se, V, and Zn) found in our study fell within the range of values usually reported for other children populations in Spain [94], Italy [55,97], and Russia [87]. However, the GM of children’s hair Fe (243 mg kg^−1^), Mn (27 mg kg^−1^), and Na (1780 mg kg^−1^) obtained in our study was around one order of magnitude higher than that found in other studies worldwide [55,87,94,97].

### 3.2. Factor and Correlation Analysis

The population characteristics are shown in Table 1. A total number of 81 children’s hair samples were collected, with a similar proportion of boys (51.1%) and girls (48.9%). Overall, 40.8% of children were aged 6–11 years and 29.6% were both <5 and 12–18-year-old. Of all children, 87.7% were underweight (BMI < 18.5).

#### 3.2.1. Gender and Age

The data obtained (Table 3, Appendix A, and Figure 6) show that both gender and age affect the levels of toxic elements in children.

Gender-related differences were found for all elements excluding Ba, Ca, Cr, Cu, Hg, Sb, and Se, with higher concentrations found especially in females (Table 3). Element concentrations in the hair of females and males were particularly different for Be (five times higher in females than males) and for Cd, Ce, Co, Cs, Fe, Ga, K, La, V, and Zr (three times higher in females than in males). Only Zn was higher in male children than females. The current literature also reports gender-related hair element concentration differences [22,55,94,98,99,100,101,102]. However, gender is often not adequately considered in the interpretation of hair analysis results [55]. Various factors (such as bodily growth, physiology, sexual hormones, and lifestyle) can affect boys’ and girls’ responses to exposure to chemicals in different ways [103]. Due to differences in kinetics, mode of action, or susceptibility, some toxic trace elements may affect the health of males and females differently [55]. In accordance with other authors [101,104], our results indicate that females may be exposed to toxic metals more than males. The results presented by Sanna et al. [96] suggest that hair is a reliable biomarker for determining population exposure levels to Pb pollution, and they indicate, in agreement with our study, that females tend to accumulate Pb in the hair more than males. The higher concentration of Sr and U in hair samples from females was also reported by other authors [101,105,106]. The high Sr, Pb, and U content in female hair can be considered a common gender characteristic due to the bone’s possible release during the growth period, which generally occurs earlier and faster than in boys [101,107,108]. 

In the puberty period, adolescents need a high amount of Zn to maintain their skeletal growth, and this demand is generally more prevalent in girls [109]. Our results are in agreement with a previous study by De Prisco et al. [110] that showed a higher Zn content in boys’ hair (244 ± 153 mg kg^−1^) than in that of girls (189 ± 34 mg kg^−1^). However, our data contrast with the results of several studies [50,55,85,96,97,111,112,113,114] that found that female hair contains more Zn than males. The lower content of Zn in female hair compared to male hair may be related to the higher presence of Cd in females. Cd can replace Zn in many metal-enzymes, and Zn deficiency may also be a symptom of Cd toxicity [81]. Human exposure to Cd in Africa is thought to be mainly due to tobacco smoking and the consumption of contaminated vegetables and crops [81]. High levels of Cd contamination were reported in lettuce grown by irrigation using water from the Akaki River in Ethiopia [115].

In accordance with other authors [55,85,102,116], also Ni, considered a carcinogen [117] and responsible for the most common type of allergy [16], was found in higher concentrations in the hair of females than males. However, our results are in contrast with the data obtained by Barbieri et al. [113] and Peña-Fernández et al. [50], who showed that the hair of males contained more Ni than females (females = 0.23 ± 4.89 mg kg^−1^ vs. males = 0.3 ± 6.5 mg kg^−1^; and females = 0.38 ± 0.34 mg kg^−1^ vs. males = 0.58 ± 0.34 mg kg^−1^, respectively).

The age distribution of the elements is shown in Appendix A and Figure 6. Concentrations of most elements (Be, Ce, Co, Fe, La, Li, Mo, and Na) in the 6–11 years age group were the highest and were significantly higher than those in the <5 years group. Element concentrations are reported according to gender and age group in Appendix A. Results showed significant differences in the hair of males with Be, Ce, Fe, La, and Ti levels in the age group of 6–11 years and Be, Mo, and U levels in the age group 12–18 years higher than those in the age group <5 years. A significant difference was observed for females with Na and Sb levels in the age group <5 years lower than those in the age group 6–11 and 12–18 years, and the age group 6–11 years, respectively. 

The increase in the element concentrations in children’s hair during the first years of childhood could be due to this group’s physiological characteristics. Children drink more water, eat more food, and breathe more air per unit weight than adults and have higher absorption rates [94]. These findings are also in accordance with the study by Kordas et al. [118], who demonstrated that older age in children is associated with a lower risk of element exposure. A decrease in hair element content with age may be due to increased excretion of chemicals for organ development and maturation [86,119].

#### 3.2.2. Passive Smoking

The hair of children with non-smoking parents contained higher Cu concentrations than that with smoking parents (cigarettes number ranging from 5 to 20), as shown in Appendix A. Environmental tobacco smoke also appears to affect K and Na content, but not significantly. Other authors have observed higher levels of essential elements in the hair of non-smokers than in smokers [120,121]. This topic is very interesting and, in the future, it should be studied in-depth, perhaps considering a greater number of samples.

#### 3.2.3. Other Characteristics and Dietary Habits of the Study Population 

Table 4 shows the element levels in the analyzed drinking water. Both the Blue Nile river’s treated and untreated water show a low content of major and trace elements. However, the concentrations of many elements (Ba, Be, Ca, Ce, Co, Cr, Cs, Fe, Ga, Hg, K, La, Li, Mg, Mn, Mo, Na, Ni, P, Rb, Sr, Ti, Tl, U, V, and Zr) in the hair of children who drink water from the Blue Nile river are significantly higher. These results require more in-depth analysis and highlight the need for monitoring the water quality of the Blue Nile river with periodic sampling.

The relation between independent variables (height, weight, BMI, and food consumption) and quantitative dependent variables (element concentrations) was also studied (Table 5). Considering the sociodemographic characteristics, a significant strong positive relationship (*p* < 0.01) with height, weight, and BMI was found in Cu [Spearman coefficient (r): 0.300, 0.327, and 0.397, respectively] and only with BMI in Sn (r: 0.315). 

A strong negative correlation was found between BMI and the following elements: Ca, Fe, Ga, P, Ti, and V. We analyzed the relationship between element levels and food group consumption, and we found a significant positive correlation (*p* < 0.01) between Ba and Hg levels and fish consumption (r: 0.320, and 0.427, respectively), and between Na content and cereals consumption (r: 0.470). A significant strong negative relationship (*p* < 0.01) with fish consumption was found in Zn (r: −0.342). No significant correlation with meat consumption was shown for all elements. It is known that fish is the major source of exposure to organic Hg [122]. The correlation between exposure to Hg and fish consumption is widely reported in the literature [57,58,59] and confirmed in the present study. Consistent with the results of the Spearman correlation test previously discussed, the children who consumed fish showed significant higher levels of Hg (GM = 0.078 mg kg^−1^, *p* < 0.001) and Ba (GM = 14 mg kg^−1^, *p* < 0.01) and lower content of Zn (GM = 76 mg kg^−1^, *p* < 0.01) than the other children (GM = 0.034, 7, and 105 mg kg^−1^ for Hg, Ba, and Zn, respectively) (Appendix A).

### 3.3. Predictors of Exposure

A multiple linear regression analysis was used to study the predictors of children’s exposure to both toxic and essential trace elements (Table 6 and Appendix A). 

Gender, age, BMI, passive smoking, drinking water, fish consumption, and fruit and vegetable consumption were significant predictors of exposure to several study population elements. The main predictor of Hg exposure in children’s hair was fish consumption (Table 6). The potential effect of age on Hg exposure is disputed, as several studies showed an increase in hair Hg with age [58,93,123,124], while many others did not observe any influence [125]. Consistent with the results of the Kruskal–Wallis test (Appendix A, and Figure 6), age is also a significant predictor of Mo and Na. 

Drinking water appears to be the major exposure predictor of some toxic elements such as Ba, Be, Cs, Li, Ni, Tl, and U (Table 6) and essential and trace elements Ca, Ce, Co, Cr, Fe, Ga, La, Mg, Mn, Mo, P, Rb, Sr, Ti, V, and Zr (Appendix A). Additional food and environmental monitoring are needed to determine the different exposure sources among children, causing differences in hair elements’ concentrations. The data currently available on the elements found in drinking water do not justify these differences. According to the results of the Mann–Whitney analysis discussed above, gender is a significant predictor of Be, Cd, Li, and U levels (Table 6) and Ca, Ce, Co, Fe, Ga, K, La, Mn, Mo, NA, P, Rb, Sr, Ti, V, Zn, and Zr (Appendix A).

### 3.4. Study Limitations

This study has some limitations. First of all, the group’s size was small (81 children), and the present study is a cross-sectional study; therefore, it does not allow an evaluation over time. 

## 4. Conclusions

Children represent a population that is particularly vulnerable to the developmental and neurotoxic effects of toxic elements and should be given special attention in biomonitoring programs.

We want to highlight the highest Pb concentrations in the hair of Ethiopian children studied compared to other literature data. A more in-depth study of this relevant finding is needed, combining different types of environmental monitoring and more extensive biomonitoring programs. The data obtained confirm gender and age as important key factors that must be taken into account in interpreting the hair analysis. The gender-based variations suggest that females are likely at greater risk for toxic element exposure than males. Dietary habits affect the elemental composition of hair in children. In particular, the hair of children who consumed fish and drank Blue Nile water contained higher Hg levels and other toxic elements (Ba, Be, Cs, Hg, Li, Ni, Tl, and U), respectively.

## Figures and Tables

**Figure 1 ijerph-17-08652-f001:**
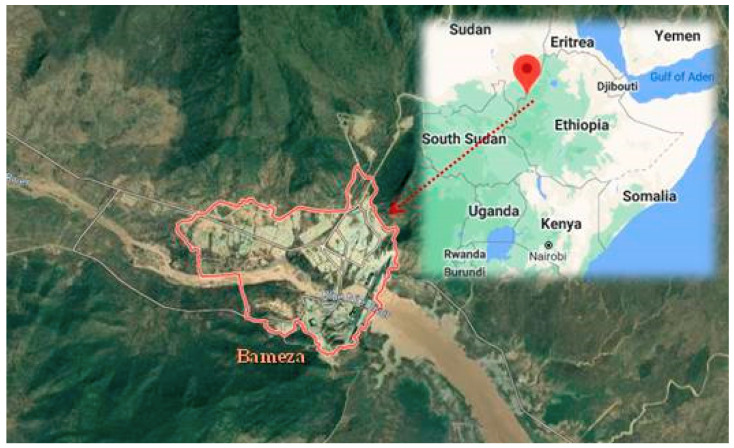
Map of the studied area (Bameza in the Benishangul-Gumuz region along the Blue Nile in north-western Ethiopia, Africa).

**Figure 2 ijerph-17-08652-f002:**
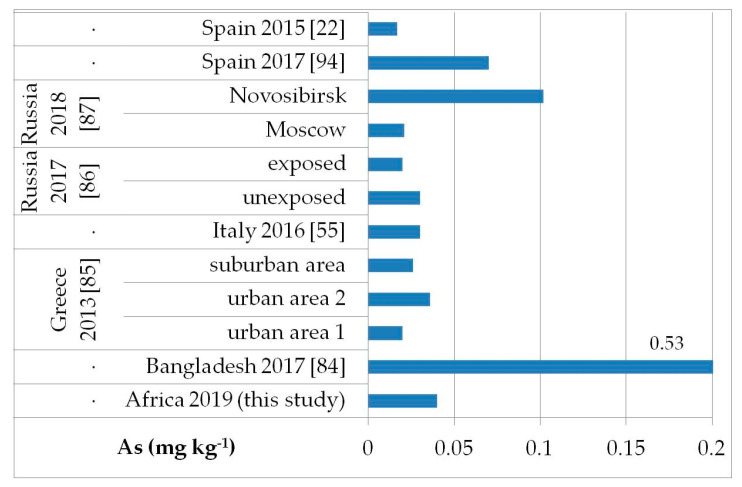
Comparison of the As concentrations in children’s hair (as geometric mean; mg kg^−1^) found in our study and other biomonitoring studies.

**Figure 3 ijerph-17-08652-f003:**
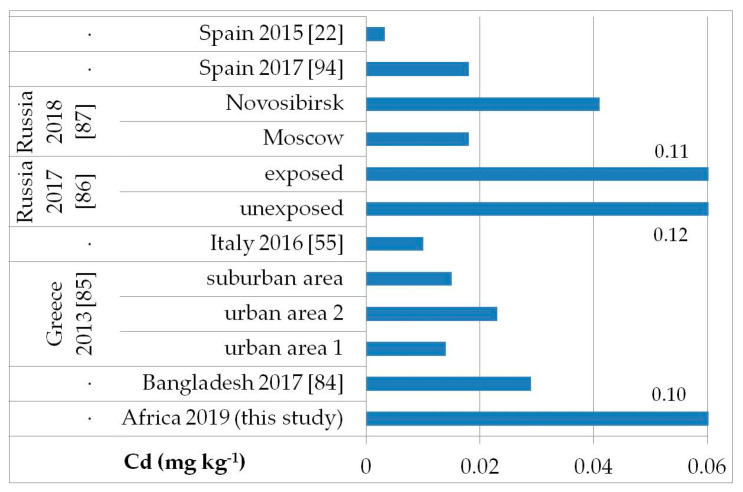
Comparison of the Cd concentrations in children’s hair (as geometric mean; mg kg^−1^) found in our study and other biomonitoring studies.

**Figure 4 ijerph-17-08652-f004:**
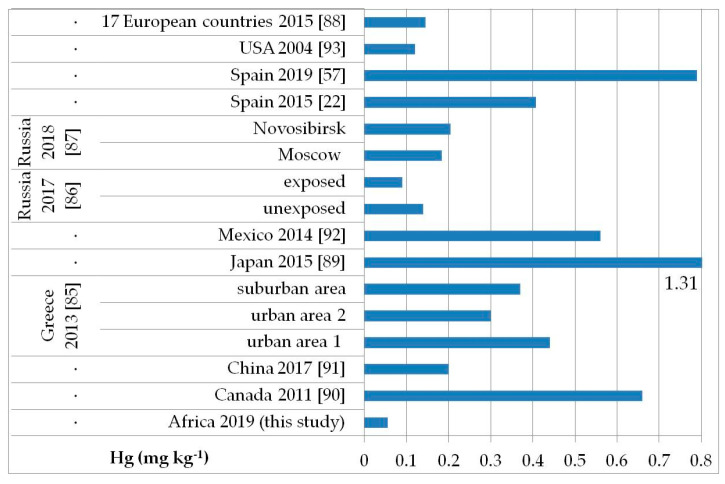
Comparison of the Hg concentrations in children’s hair (as geometric mean; mg kg^−1^) found in our study and other biomonitoring studies.

**Figure 5 ijerph-17-08652-f005:**
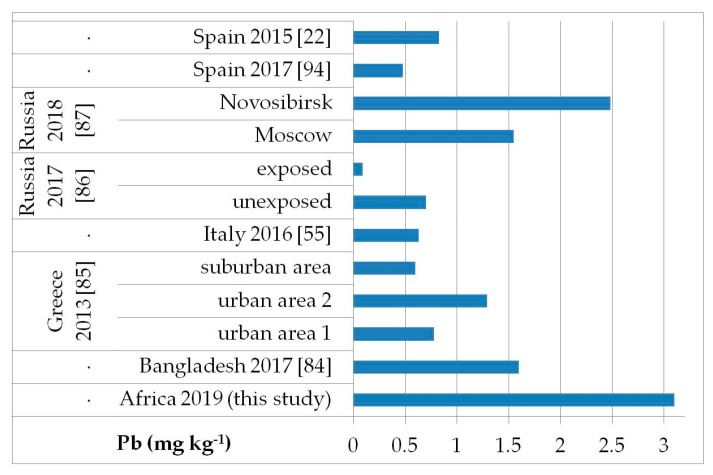
Comparison of the Pb concentrations in children’s hair (as geometric mean; mg kg^−1^) found in our study and other biomonitoring studies.

**Figure 6 ijerph-17-08652-f006:**
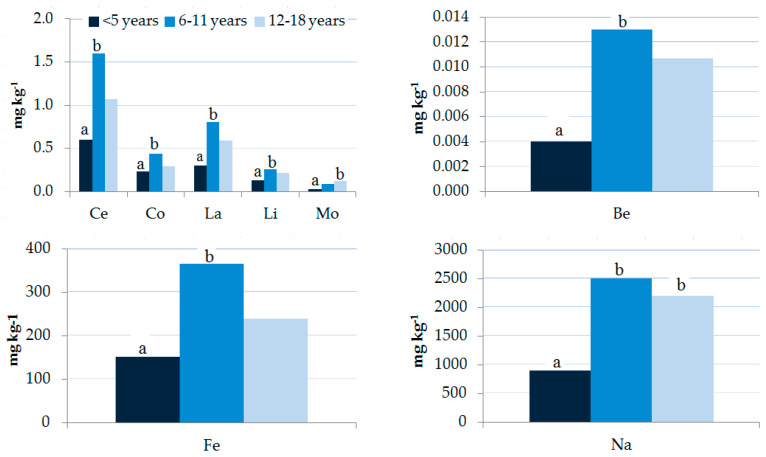
Influence of age on element levels (geometric mean, mg kg^−1^) in children’s hair samples. A non-parametric Kruskal–Wallis test was applied. Different letters “a” and “b” for the same element show significant differences (*p* < 0.05).

**Table 1 ijerph-17-08652-t001:** Characteristics of the studied population and food consumption.

Variable	^a^ N	^b^ N%	^c^ Median (min–max)
**Studied population characteristics**	81	-	-
**Gender**	81	100	-
Male	40	49.4	-
Female	41	50.6	-
**Height** (**cm**)	79	97.5	127 (60–181)
**Weight** (**kg**)	80	98.8	22 (4–60)
**Body mass index** (**kg m^−2^**)	79	97.5	15.0 (9.5–26)
Underweight (<18.5)	71	87.7	-
Normal weight (18.5–24.9)	7	8.6	-
Overweight/obesity (>25)	1	1.2	-
**Age** (**years**)	81	100	9 (4 months–18 years)
< 5 years	24	29.6	-
6–11 years	33	40.8	-
12–18 years	24	29.6	-
**Passive smoking**	77	95.1	-
Yes	14	17.3	-
No	63	77.8	-
**Food consumption (g month^−1^ or mL month^−1^)**			
**Water or breast milk**	81	100	
Bottled and/or treated water of the Blue Nile river	40	49.4	28,000 (14,000–84,000)
The water of the Blue Nile river	33	40.7	28,000 (14,000–56,000)
Breast milk	8	9.9	30,800 (28,000–42,000)
**Meat** (**bovine, goat, chicken**)	80	98.8	-
Yes	68	84.0	800 (100–2000)
No	12	14.8	-
**Fish**	80	98.8	-
Yes	47	58.0	800 (400–2000)
No	33	40.7	-
**Fish** (**caught in the Blue Nile river**)	43	53.1	400 (400–2000)
**Canned** (**tuna, salmon**) **and caught fish in the Blue Nile river**	4	4.9	600 (400–800)
**Cereals** (**wheat, rice, oats, teff**)	67	82.7	-
Yes	61	75.3	4480 (1120–4480)
No	6	7.4	
**Vegetables and fruit** (**orange, banana, mango, papaya**)	80	98.8	-
Yes	63	77.8	1200 (100–4800)
No	17	21	

^a^ N = data number. ^b^ N% = data number percentage. ^c^ Median (min–max) = median (minimum–maximum range).

**Table 2 ijerph-17-08652-t002:** Element levels (mg kg^−1^) in the hair of children living in the Benishangul-Gumuz region in Ethiopia (*n* = 81).

Element	^a^ DL	% <DL	^b^ GM	^c^ AM	^d^ SD	^e^ Min	Percentile				^f^ Max
							25th	50th	75th	95th	
Al	7 × 10^−2^	40.7	1	25	44	<DL	<DL	6	42	127	233
As	2 × 10^−2^	35.8	0.04	0.08	0.10	<DL	<DL	0.04	0.09	0.26	0.65
B	2 × 10^0^	79	<DL	<DL	-	<DL	<DL	<DL	<DL	4.6	29.2
Ba	5 × 10^−1^	1.2	11	18	18	<DL	7	14	20	59	74
Be	2 × 10^−4^	9.9	0.008	0.027	0.041	<DL	0.004	0.012	0.027	0.149	0.181
Bi	2 × 10^−4^	44.4	0.001	0.013	0.063	<DL	<DL	0.001	0.007	0.035	0.551
Ca	1 × 10^+1^	2.5	1480	2630	3000	<DL	916	1610	2810	8960	17,300
Cd	1 × 10^−4^	1.2	0.10	0.18	0.26	<DL	0.06	0.10	0.21	0.75	1.58
Ce	6 × 10^−4^	0	1.0	2.1	2.7	0.03	0.5	1.2	2.2	8.7	13.4
Co	1 × 10^−3^	0	0.32	0.60	0.80	0.02	0.16	0.32	0.57	3.09	3.58
Cr	6 × 10^−3^	1.2	0.6	1.1	1.5	<DL	0.3	0.5	1.1	4.9	6.5
Cs	4 × 10^−4^	1.2	0.013	0.027	0.035	<DL	0.007	0.015	0.027	0.112	0.156
Cu	1 × 10^−2^	0	11.5	12.3	5.1	4.9	8.9	11.2	13.5	25.3	32.7
Fe	7 × 10^−2^	0	243	585	890	11	96	250	585	2980	4000
Ga	7 × 10^−4^	0	0.07	0.16	0.23	0.01	0.03	0.08	0.15	0.84	1.07
Hg	4 × 10^−3^	0	0.056	0.085	0.089	0.012	0.026	0.053	0.111	0.273	0.483
K	3 × 10^+1^	2.5	1060	1900	1700	<DL	513	1410	3100	5420	6690
La	2 × 10^−4^	0	0.6	1.1	1.4	0.01	0.3	0.6	1.1	4.7	7.3
Li	4 × 10^−3^	0	0.19	0.32	0.33	0.02	0.09	0.2	0.38	1.16	1.6
Mg	3 × 10^0^	0	277	394	400	30	168	272	429	1409	2320
Mn	1 × 10^−2^	0	27	44	41	1	14	36	55	161	184
Mo	2 × 10^−3^	7.4	0.07	0.16	0.22	<DL	0.04	0.12	0.19	0.53	1.53
Na	4 × 10^0^	0	1780	2850	2900	160	819	1970	3780	9070	15,000
Nb	1 × 10^−4^	65.4	0.001	0.08	0.19	<DL	<DL	<DL	0.02	0.64	0.88
Ni	3 × 10^−3^	0	1.2	1.8	1.8	0.2	0.7	1.1	2.0	6.6	8.8
P	3 × 10^0^	2.5	98	129	100	<DL	79	97	147	406	611
Pb	5 × 10^−3^	0	3.1	4.8	5.5	0.3	1.7	3.0	5.4	15.5	32.8
Rb	2 × 10^−3^	0	1.3	2.0	1.8	0.1	0.7	1.6	2.8	5.6	8.6
Sb	8 × 10^−4^	6.2	0.07	0.23	0.99	<DL	0.05	0.10	0.15	0.36	8.85
Se	5 × 10^−2^	4.9	0.19	0.23	0.13	<DL	0.15	0.20	0.28	0.58	0.67
Si	5 × 10^0^	23.5	65	167	150	<DL	10	156	258	466	499
Sn	3 × 10^−4^	1.2	0.23	0.35	0.35	<DL	0.13	0.26	0.42	0.98	2.38
Sr	3 × 10^−2^	0	19	25	19	2	11	22	33	66	99
Te	2 × 10^−3^	66.7	<DL	<DL	-	<DL	<DL	<DL	<DL	0.0078	0.0148
Ti	2 × 10^−2^	0	14	31	49	2	6	12	31	171	253
Tl	5 × 10^−5^	0	0.0054	0.0068	0.0053	0.0007	0.0035	0.0054	0.0076	0.0217	0.0262
U	2 × 10^−5^	0	0.029	0.046	0.045	0.002	0.014	0.031	0.057	0.165	0.204
V	4 × 10^−3^	1.2	0.8	1.8	2.7	<DL	0.3	0.8	1.5	10.2	12.0
W	8 × 10^−4^	86.4	0.0006	0.0019	0.0061	<DL	<DL	<DL	<DL	0.0129	0.0462
Zn	2 × 10^−1^	2.5	86	117	70	<DL	77	100	152	245	451
Zr	7 × 10^−4^	0	0.39	0.89	1.31	0.01	0.16	0.37	0.99	4.59	6.29

^a^ DL = detection limit. ^b^ GM = geometric mean. ^c^ AM = arithmetic mean. ^d^ SD = standard deviation. ^e^ Min = minimum value. ^f^ Max = maximum value.

**Table 3 ijerph-17-08652-t003:** Gender influence on element levels (mg kg^−1^) in children’s hair.

Element	Males			Females			^d^*p* Value
	^a^ GM	^b^ Min	^c^ Max	^a^ GM	^b^ Min	^c^ Max	
Al	2	<DL	233	1	<DL	167	-
As	0.04	<DL	0.65	0.04	<DL	0.32	-
B	<DL	<DL	5.1	1.4	<DL	29.2	-
Ba	8	<DL	74	15	1	73	ns
Be	0.004	<DL	0.065	0.019	0.001	0.181	***
Bi	<0.0014	<DL	0.076	0.002	<DL	0.551	-
Ca	1080	<DL	7570	2110	284	17,300	ns
Cd	0.06	<DL	0.34	0.16	0.03	1.58	***
Ce	0.6	0.03	6.2	1.8	0.2	13.4	***
Co	0.20	0.02	1.38	0.52	0.1	3.58	***
Cr	0.5	<DL	6.5	0.8	0.1	6.1	ns
Cs	0.008	<DL	0.085	0.021	<DL	0.156	**
Cu	10.8	5.2	25.8	12.2	4.9	32.7	ns
Fe	149	11	1620	419	29	4000	**
Ga	0.05	0.005	0.48	0.12	0.01	1.07	**
Hg	0.056	0.013	0.367	0.058	0.012	0.483	ns
K	639	<DL	4420	1750	178	6690	***
La	0.3	0.01	3.3	1.0	0.1	7.3	***
Li	0.14	0.02	1.43	0.28	0.02	1.60	**
Mg	207	44	746	391	114	2320	***
Mn	18	1	71	45	3	184	***
Mo	0.05	<DL	0.68	0.12	<DL	1.53	***
Na	1290	160	10,200	2500	247	15000	**
Nb	<0.0013	<DL	0.17	0.003	<DL	0.88	-
Ni	1.0	0.2	8.8	1.5	0.3	6.9	*
*p*	76	2	239	126	21	611	*
Pb	2.5	0.3	32.8	4.1	0.9	26.6	*
Rb	0.9	0.1	4.5	1.9	0.3	8.6	**
Sb	0.05	<DL	8.85	0.10	0.01	0.49	ns
Se	0.19	<DL	0.59	0.19	<DL	0.67	ns
Si	45	<DL	499	100	<DL	484	-
Sn	0.18	<DL	2.38	0.30	0.04	1.01	*
Sr	15	3	44	24	5	99	*
Te	<DL	<DL	0.0077	<DL	<DL	0.0148	-
Ti	10	2	76	22	2	253	**
Tl	0.0046	0.0018	0.0126	0.0065	<0.0017	0.0262	**
U	0.020	0.002	0.165	0.043	0.006	0.204	**
V	0.5	<DL	4.1	1.4	0.2	12.0	***
W	<DL	<DL	0.0128	<DL	<DL	0.0462	-
Zn	96	<DL	451	78	<DL	251	*
Zr	0.24	0.01	1.90	0.66	0.02	6.29	**

^a^ GM = geometric mean. ^b^ Min = minimum value. ^c^ Max = maximum value. ^d^ Mann–Whitney test: ns = not significant at *p* > 0.05; and significant at *p* < 0.05 (*), *p* < 0.01 (**), and *p* < 0.001 (***). Elements with a detection frequency percentage >detection limit (DL) lower than 80% were excluded from the statistical calculation: Al, As, B, Bi, Nb, Si, Te, and W.

**Table 4 ijerph-17-08652-t004:** Element concentrations in drinking water (μg L^−1^) and in children’s hair (mg kg^−1^) according to water consumption.

Element	^a^ DL in Water	Drinking Water				Children Hair						
		Treated Water		Blue Nile River Water		Bottled and/or Treated Water			Blue Nile River			^g^ P
		^b^ AM	^c^ SD	^b^ AM	^c^ SD	^d^ GM	^e^ Min	^f^ Max	^d^ GM	^e^ Min	^f^ Max	
Al	0.1	0.26	0.09	0.7	2.0	3	<DL	233	0.4	<DL	70	-
As	0.1	0.12	0.07	0.24	0.01	0.03	<DL	0.65	0.06	<DL	0.32	-
B	4	9.17	0.13	11.5	0.1	<DL	<DL	<DL	1.6	<DL	29.2	-
Ba	0.7	18.5	0.7	11.8	0.7	7	<DL	23	24	7	74	***
Be	0.008	<DL		<DL		0.004	<DL	0.035	0.026	<DL	0.181	***
Bi	0.009	<DL		<DL		0.001	<DL	0.551	0.001	<DL	0.039	-
Ca	23	20,300	900	20,800	1500	878	<DL	3280	2760	580	17,300	***
Cd	0.001	0.0374	0.0001	0.0061	0.0048	0.12	<DL	1.58	0.09	0.01	0.56	ns
Ce	0.004	0.011	0.002	0.11	0.12	0.6	0.03	3.4	2.3	0.3	12.9	***
Co	0.007	0.097	0.003	0.11	0.03	0.20	0.02	0.81	0.69	0.14	3.58	***
Cr	0.006	0.044	0.002	0.124	0.025	0.4	<DL	6.5	1.0	0.1	5.2	***
Cs	0.0005	0.0185	0.0011	0.0026	0.0011	0.007	<DL	0.061	0.031	0.005	0.150	***
Cu	0.02	1.23	0.08	1.61	0.08	12.9	6.5	32.7	10.3	4.9	16.0	-
Fe	2	<DL		34	42	129	11	578	630	59	4000	***
Ga	0.0003	0.0010	0.0009	0.022	0.025	0.04	0.01	0.20	0.17	0.02	1.07	***
Hg	0.003	0.0530	0.0057	0.0475	0.0035	0.043	0.012	0.367	0.089	0.018	0.483	***
K	20	2070	11	1860	110	742	<DL	5400	1770	155	6690	**
La	0.0006	0.0022	0.0001	0.045	0.052	0.3	0.01	1.9	1.2	0.1	6.6	***
Li	0.005	1.43	0.01	0.0769	0.0029	0.15	0.02	0.60	0.33	0.05	1.60	***
Mg	0.5	5811	61	6210	176	201	44	746	451	140	2320	***
Mn	0.07	9.6	0.7	1.5	2.0	19	2	81	52	12	184	***
Mo	0.2	0.644	0.006	0.688	0.015	0.05	<DL	0.89	0.15	<DL	1.53	***
Na	2	7760	37	7750	181	1810	160	15,000	2180	180	10,200	ns
Nb	0.01	<DL		0.024	0.020	0.0002	<DL	0.04	0.01	<DL	0.88	-
Ni	0.04	0.93	0.36	0.728	0.067	0.9	0.2	4.2	1.8	0.2	8.8	**
P	2	38	4	82	16	72	<DL	407	139	28	611	***
Pb	0.001	0.065	0.021	0.039	0.012	3.8	0.5	32.8	2.5	0.3	9.1	**
Rb	0.001	1.79	0.01	0.985	0.003	1.0	0.1	4.5	2.1	0.4	8.6	**
Sb	0.003	0.0281	0.0028	0.0219	0.0062	0.05	<DL	0.62	0.09	<DL	0.49	ns
Se	0.06	0.072	0.042	0.0899	0.0040	0.18	<DL	0.59	0.19	0.06	0.58	ns
Si	33	2660	13	6140	376	38	<DL	333	197	<DL	499	-
Sn	0.006	0.0133	0.0066	0.0146	0.0032	0.28	<DL	2.38	0.18	0.04	0.95	**
Sr	0.05	131	2	121	2	14	3	44	30	7	99	***
Te	0.02	0.0041	0.0055	0.0045	0.0047	0.0012	<DL	0.0148	0.0016	0.0008	0.0092	-
Ti	0.01	0.879	0.005	2.2	2.1	8	2	31	33	4	253	***
Tl	0.0002	0.00880	0.00042	0.00515	0.00078	0.0043	0.0007	0.0216	0.0074	0.0023	0.0262	***
U	0.0006	0.010	0.004	0.079	0.014	0.020	0.002	0.165	0.051	0.010	0.204	***
V	0.04	1.04	0.05	2.40	0.20	0.4	<DL	1.4	2.0	0.3	12.0	***
W	0.06	<DL		<DL		0.0005	<DL	0.0462	<DL	<DL	0.0210	-
Zn	0.07	2.68	0.52	0.29	0.25	95	<DL	451	71	10	202	***
Zr	0.004	<DL		0.111	0.093	0.21	0.01	2.03	0.97	0.08	6.29	***

^a^ DL = detection limit. ^b^ AM = arithmetic mean. ^c^ SD = standard deviation. ^d^ GM = geometric mean. ^e^ Min = minimum value. ^f^ Max = maximum value. ^g^ Mann–Whitney test: ns = not significant at *p* > 0.05; and significant at *p* < 0.01 (**) and *p* < 0.001 (***). Elements with a detection frequency percentage >DL lower than 80% were excluded from the statistical calculation: Al, As, B, Bi, Nb, Si, Te, and W.

**Table 5 ijerph-17-08652-t005:** Correlations between the levels of the elements in the hair and the characteristics of children.

^a^ Element	Sociodemographic Characteristics			Food Consumption (g Month^−1^ or mL Month^−1^)				
	Height (cm)	Wight (kg)	^b^ BMI (kg m^−2^)	Breast Milk	Meat	Fish	Cereals	Fruit and Vegetables
B	−0.091	−0.132	−0.241 *	0.094	−0.181	0.134	0.067	−0.128
Ba	0.131	0.072	−0.282 *	−0.073	−0.119	0.320 **	0.129	−0.200
Be	0.180	0.106	−0.291 *	−0.132	−0.049	0.199	0.228	−0.076
Ca	−0.003	−0.077	−0.317 **	0.120	−0.215	0.108	−0.083	−0.242 *
Cd	0.206	0.197	0.178	−0.207	0.116	−0.167	0.108	0.158
Ce	0.141	0.083	−0.265 *	−0.094	−0.070	0.241 *	0.263 *	−0.199
Co	0.083	0.021	−0.280 *	−0.116	−0.058	0.229 *	0.231	−0.242 *
Cr	−0.019	−0.055	−0.240 *	0.053	−0.132	0.193	−0.028	−0.110
Cs	0.138	0.074	−0.276 *	−0.016	−0.154	0.194	0.183	−0.174
Cu	0.300 **	0.327 **	0.393 **	−0.149	0.086	−0.250 *	0.313 *	0.160
Fe	0.100	0.034	−0.334 **	−0.064	−0.119	0.224	0.250	−0.251 *
Ga	0.089	0.031	−0.299 **	−0.074	−0.120	0.222	0.228	−0.224
Hg	−0.008	0.018	−0.125	−0.169	0.128	0.427 **	0.009	−0.129
K	0.146	0.119	−0.083	−0.034	−0.112	0.109	0.315 *	−0.160
La	0.140	0.084	−0.242 *	−0.086	−0.093	0.239 **	0.250	−0.198
Li	0.138	0.107	−0.219	−0.086	−0.084	0.090	0.233	−0.170
Mg	0.149	0.087	−0.204	−0.027	−0.098	0.156	0.208	−0.151
Mn	0.089	0.029	−0.261 *	−0.081	−0.044	0.139	0.184	−0.218
Mo	0.213	0.164	−0.189	−0.159	−0.029	0.213	0.278 *	−0.081
Na	0.274 *	0.279 *	0.072	−0.218	0.109	−0.008	0.470 **	−0.032
Ni	0.024	−0.012	−0.187	−0.067	0.040	0.013	0.110	−0.055
P	0.091	0.016	−0.293 *	−0.068	−0.071	0.214	0.182	−0.109
Pb	−0.042	−0.023	0.159	−0.085	0.066	−0.258 **	0.002	0.175
Rb	0.116	0.095	−0.111	−0.013	−0.130	0.112	0.257 *	−0.164
Sb	−0.190	−0.236 *	−0.118	0.227 *	−0.179	−0.051	−0.282 **	−0.148
Se	−0.199	−0.134	0.027	0.152	−0.001	−0.127	−0.222	−0.173
Sn	−0.002	0.038	0.315 **	−0.088	0.154	−0.194	−0.146	0.217
Sr	0.121	0.072	−0.220	0.022	−0.113	0.185	0.098	−0.254 *
Ti	0.046	0.004	−0.301 **	−0.062	−0.107	0.266 **	0.230	−0.268 *
Tl	0.059	0.039	−0.228 *	0.004	−0.050	0.184	0.210	−0.129
U	0.133	0.113	−0.133	−0.037	−0.145	0.202	0.236	−0.244 *
V	0.077	0.019	−0.326 **	−0.120	−0.042	0.237 *	0.238	−0.219
Zn	0.074	0.062	0.205	−0.003	0.142	−0.342 **	−0.134	0.201
Zr	−0.012	−0.046	−0.268 *	−0.096	0.005	0.242 **	0.129	−0.214

^a^ Elements (Al, As, B, Bi, Nb, Si, Te, and W) with a detection frequency percentage > detection limit lower than 80% were excluded from the statistical calculation. Rho Spearman correlation test is significant for *p* < 0.05 (*) and *p* < 0.01 (**). ^b^ BMI = body mass index.

**Table 6 ijerph-17-08652-t006:** Results of the backward multiple linear regression model analysis for some toxic elements.

Elements	Factors	^a^ B	^b^ SE	^c^ β	*p* Value	95% ^d^ CI for B		R^2^	Adjusted R^2^
						Lower Bound	Upper Bound		
Ba	Constant	2.85	0.302		<0.001	2.24	3.46	0.501	0.479
	Fruit and vegetables	−0.925	0.294	−0.334	0.003	−1.52	−0.333		
	Drinking water	1.07	0.194	0.583	<0.001	0.676	1.46		
Be	Constant	−3.34	0.430		<0.001	−4.21	−2.47	0.587	0.545
	Gender	−0.683	0.235	−0.299	0.006	−1.16	−0.207		
	Fish	−0.726	0.323	−0.292	0.030	−1.38	−0.073		
	Fruit and vegetables	−1.05	0.353	−0.318	0.005	−1.77	−0.340		
	Drinking water	1.68	0.287	0.738	<0.001	1.10	2.26		
Cd	Constant	−1.63	0.194		<0.001	−2.02	−1.24	0.233	0.199
	Gender	−0.581	0.227	−0.337	0.014	−1.04	−0.124		
Cs	Constant	−3.74	0.411		<0.001	−4.57	−2.91	0.435	0.396
	Fruit and vegetables	−0.851	0.377	−0.261	0.029	−1.61	−0.091		
	Drinking water	1.14	0.252	0.523	<0.001	0.632	1.65		
Hg	Constant	−2.61	0.338		<0.001	−3.289	−1.929	0.339	0.295
	Age	−0.362	0.151	−0.293	0.021	−0.667	−0.057		
	Fish	0.873	0.253	0.423	0.001	0.363	1.38		
Li	Constant	−1.18	0.294		<0.001	−1.77	−0.584	0.339	0.295
	Gender	−0.518	0.236	−0.270	0.033	−0.993	−0.043		
	Drinking water	0.577	0.248	0.300	0.025	0.077	1.08		
Ni	Constant	0.947	0.375		0.015	0.192	1.70	0.252	0.202
	Fruit and vegetables	−0.797	0.325	−0.326	0.018	−1.45	−0.142		
	Drinking water	0.757	0.250	0.473	0.004	0.254	1.26		
Sn	Constant	−0.992	0.199		<0.001	−1.39	−0.591	0.314	0.267
	Drinking water	−0.757	0.261	−0.442	0.006	−1.28	−0.230		
Tl	Constant	−5.44	0.297		<0.001	−6.04	−4.84	0.345	0.301
	Fruit and vegetables	−0.660	0.243	−0.341	0.009	−1.15	−0.171		
	Drinking water	0.446	0.157	0.351	0.007	0.129	0.763		
U	Constant	−3.24	0.283		<0.001	−3.81	−2.67	0.412	0.373
	Gender	−0.478	0.227	−0.243	0.041	−0.936	−0.020		
	Drinking water	0.761	0.239	0.387	0.003	0.279	1.24		

^a^ B = non-standardized regression coefficients. ^b^ SE = standard error. ^c^ β = standardized regression coefficients. ^d^ CI = confidence interval. No variables included in the models were processed for Al, As, B, Bi, Nb, Pb, Sb, Si, Te, and W.

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
