# Peer review of "Element Levels and Predictors of Exposure in the Hair of Ethiopian Children"

_ijerph, 2020, doi:10.3390/ijerph17228652_

Round 1

Reviewer 1 Report

The article “ Element levels and predictors of exposure in the hair of Ethiopian children” reports interesting articles that are novel for this study and there is lack of studies for that area of the world. However, there are some issues that may improve the paper. I have pointed out some of my concerns that mau help authors improving their article.

Lines 63-65: I suggest to remove all the elements in parenthesis and instead of toxic elements write toxic heavy metals

Line 66: Remove” (Africa) [51].”

Line 43, I suggest to use these references that talks about dietary exposure of heavy metals:

https://doi.org/10.1021/acssuschemeng.9b03531

https://doi.org/10.1016/j.ecoenv.2020.110177

Line75:Instead of the “aged 0–18 years” write “aged under18”

Figure 1, it is better to show the location of the study in reference to the geographical location of the country

95- The sample collection needs more detain. For example, replicates, deph of sampling. What the authors mean by improved water?

L 103 and 104 remove the “ones such as,” and “ such as” and replace it by “of”

119: replace the “to 95 °C” with “at 95 °C” and “to cool” to “ to cool down”

123: using the duplicates instead of triplicates in analysis raised my concern over the statistical analysis that has been processed. I am not sure how it may be reliable with the duplicate samples.

In  Table1 and other places instead of the “0–5 years “, please write the “<5 years”

In Table 1, there is no point  to count for the missing data (%), I suggest that author recalculate based on existing data. Also, for the Height (cm) and Weight (kg) remove the missing data. What is the point of missing data for the “Water or breast milk “? Authors should reorganize the Table 1 in three different columns at least or more, n, %, min-Max, , Yes, No etc

Authors need to clarify what BMI stand for in this table

Figure 3, if it is possible add the standard deviation error bars on top of each column. Is it done as duplicates?

Table 5, in this context I suggest to report only positive numbers. There is no point to show he negative correlation. Authors may show the negative values as NA or nothing(-_)

Author Response

Reviewer 1 (R1)

Comments and Suggestions for Authors

The article “Element levels and predictors of exposure in the hair of Ethiopian children” reports interesting articles that are novel for this study and there is lack of studies for that area of the world. However, there are some issues that may improve the paper. I have pointed out some of my concerns that mau help authors improving their article.

Lines 63-65: I suggest to remove all the elements in parenthesis and instead of toxic elements write toxic heavy metals

Authors (A): Ok, done. Thank you for your kind review report.

R1: Line 66: Remove” (Africa) [51].”

A: Done.

R1: Line 43, I suggest to use these references that talks about dietary exposure of heavy metals:

https://doi.org/10.1021/acssuschemeng.9b03531

https://doi.org/10.1016/j.ecoenv.2020.110177

A: Done, thank you for your suggestion.

R1: Line75:Instead of the “aged 0–18 years” write “aged under18”

A: Done

R1: Figure 1, it is better to show the location of the study in reference to the geographical location of the country

A: Done, thank you for your suggestion.

R1: 95- The sample collection needs more detain. For example, replicates, deph of sampling. What the authors mean by improved water?

A: Thank you for your comment. Further details on water sampling were added in section 2.3.2., following your suggestion and that of Reviewer 3. The word “improved” was replaced with "treated". The water of the Blue Nile River is treated to make it safe to drink.

R1: L 103 and 104 remove the “ones such as,” and “ such as” and replace it by “of”

119: replace the “to 95 °C” with “at 95 °C” and “to cool” to “ to cool down”

A: Done, thank you.

R1: 123: using the duplicates instead of triplicates in analysis raised my concern over the statistical analysis that has been processed. I am not sure how it may be reliable with the duplicate samples.

A: Thank you for your comment. The methods used to determine the elemental content in human hair were validated in previous studies (Astolfi, M. L.; Protano, C.; Marconi, E.; Massimi, L.; Brunori, M.; Piamonti, D.; Migliara G.; Vitali, M.; Canepari, S. A new rapid treatment of human hair for elemental determination by inductively coupled mass spectrometry. Anal. Methods 2020, 12, 1906–1918. https://doi.org/10.1039/C9AY01871A; Astolfi, M.L.; Protano, C.; Marconi, E.; Massimi, L.; Piamonti, D.; Brunori, M.; Vitali, M.; Canepari, S. Biomonitoring of Mercury in Hair among a Group of Eritreans (Africa). Int. J. Environ. Res. Public Health 2020, 17, 1911. https://doi.org/10.3390/ijerph17061911; Astolfi, M. L.; Protano, C.; Marconi, E.; Piamonti, D.; Massimi, L.; Brunori, M.; Vitali, M.; Canepari, S. Simple and rapid method for the determination of mercury in human hair by cold vapour generation atomic fluorescence spectrometry. Microchem. J. 2019, 150, 104186. https://doi.org/10.1016/j.microc.2019.104186). All analytical merit figures are known, including the accuracy of the method.

R1: In  Table1 and other places instead of the “0–5 years “, please write the “<5 years”

A: Done, thank you.

R1: In Table 1, there is no point  to count for the missing data (%), I suggest that author recalculate based on existing data. Also, for the Height (cm) and Weight (kg) remove the missing data. What is the point of missing data for the “Water or breast milk “? Authors should reorganize the Table 1 in three different columns at least or more, n, %, min-Max, , Yes, No etc

Authors need to clarify what BMI stand for in this table.

A: Done.

R1: Figure 3, if it is possible add the standard deviation error bars on top of each column. Is it done as duplicates?

A: Please note that Figure 3 shows the trends of mean concentration as a geometric mean (not an arithmetic mean) in the hair of the entire population of children by age group.

R1: Table 5, in this context I suggest to report only positive numbers. There is no point to show he negative correlation. Authors may show the negative values as NA or nothing(-_)

A: Thank you for your comment. The term correlation indicates the relationship between two variables or more. Please note that the relationship observed is not, generally, linked by a cause-effect relationship but represents a variable’s ability to change as a function of another. In our opinion, the negative correlations are also important, which in this specific field show that as the variable considered increases, the concentration of a specific element decreases.

Reviewer 2 Report

* hair/matrix window period - hair has a 180 days wide period of detection, at least.

* related to technology; products - the region is an industrial production site ?

* age - some metals have an accumulative profile.

It´ll be great hearing about " hair/matrix window period " or " retrospective analysis, in case of forensic.

Interesting data about Li, Cd and Ni, trace metals related to " technology " products, and age.

Congrats !

Author Response

Reviewer 2 (R2)

Comments and Suggestions for Authors

* hair/matrix window period - hair has a 180 days wide period of detection, at least.

Authors (A): Thank you for your comment. The following sentence was added to the “Introduction” section:  “Hair grows at a rate of  ~1 cm/month; therefore, the chronological exposure to elements can be traced from the segmental hair analysis over a defined window period (at least 180 days) depending on the selected hair length [52]. The hair ability to sequentially accumulate chemicals in its inner structure, together with the opportunity to conduct retrospective analyzes, mean that hair analysis can be used for screening and confirmation purposes in various application contexts such as forensic and clinical [53]”.

R2: * related to technology; products - the region is an industrial production site ?

A: The study area is rural, and its inhabitants mainly practice sedentary peasant agriculture or derive their livelihood from hunting (Getahun, B.T.; Tsega, A.D. Centre-Periphery Relations in Ethiopian Empire: The Case of Benishangul Gumuz, 1898-1941. Int. J. Human. Soc. Sci. Res.2014, 2, 1-12). However, the possible sources of exposure to the toxic heavy metals analyzed can be multiple in Ethiopia, such as contaminated food, water, and soil pollution, especially from abuse or misuse of chemicals in agriculture, indoor and outdoor air pollution from inefficient combustion of fuels solids (wood, animal dung, charcoal, crop scraps, and coal) and waste combustion, residues of industrial manufacturing products, alluvial gold mining, oil mining and refining [WHO. Chemicals of public health concern in the African Region and their management: Regional Assessment Report, World Health Organization. Regional Office for Africa, Republic of Congo, 2014. Available online: https://www.afro.who.int/sites/default/files/2017-06/9789290232810.pdf (Accessed on 31 October 2020); Addis Ababa Policy Planning Directorate, Federal Ministry of Health;. Health and Health Related Indicators. Version 1.2008 E.C., 2015. Available online: https://www.dktethiopia.org/sites/default/files/PublicationFiles/Health%20and%20Health%20Related%20Indicator%20 2008.pdf (accessed on 31 October 2020); WHO: Global Health Estimates. World Health Organization, Geneva, Switzerland, 2016. Available online:  http://www. who.int/healthinfo/global_burden_disease/estimates/en (Accessed 9 August 2018)]. These topics were added and further described in the "Results and Discussion" section.

R2: * age - some metals have an accumulative profile.

A: Thank you for your comment. We agree with you and discussed the variability among different age groups (3.2.1 section).

R2: It´ll be great hearing about " hair/matrix window period " or " retrospective analysis, in case of forensic.

A: Thank you for your suggestion. We emphasized the importance and usefulness of segmental hair analysis in the “Introduction” section. Please note that the hair was very short in our case, and the data can be referred to as a time window of about one month.

R2: Interesting data about Li, Cd and Ni, trace metals related to " technology " products, and age.

Congrats !

A: Thank you very much.

Reviewer 3 Report

The manuscript reports a clinical toxicology study that demonstrated the concentrations of elements in the hair of children living in a rural area of Ethiopia (Africa).

Nevertheless, I have some comments:

  1. The authors should provide further details about the study region. Is this an urban region? rural? What are the possible sources of exposure to these toxic metals?
  2. The authors must inform about the water samples collected in the region. How was the collection, what was the volume collected, how was the transportation of these samples? Describe how the water was sampled (e.g., what were the methods and apparatus applied?), how were samples transported and preserved (i.e., acidified? Frozen?)
  3. The authors must cite the type of water sample that was collected and the number of samples.
  4. The abstract must be rewritten to represent the study results accordingly.
  5. What care is taken in collecting hair samples to avoid external contamination by elements?
  6. The authors conclude that it is essential to assess children's element levels due to possible neurotoxicity involvement. In this context, the authors could be neuropsychological assessments with the study group.

Author Response

Reviewer 3 (R3)

Comments and Suggestions for Authors

The manuscript reports a clinical toxicology study that demonstrated the concentrations of elements in the hair of children living in a rural area of Ethiopia (Africa).

Nevertheless, I have some comments:

  1. The authors should provide further details about the study region. Is this an urban region? rural? What are the possible sources of exposure to these toxic metals?

Authors (A): First of all, thank you for your kind review report. The study area is rural, as described on page 2, line 75. The following further details of the study area were added to the revised text: “A cross-sectional study was conducted between November and December 2019 in Bameza, a rural area of the Benishangul-Gumuz region along the Blue Nile river in north-western Ethiopia (Africa) (Figure 1). Bameza is approximately 150 km from Binshangul Gumuz's capital Asosa and 500 km from Ethiopia's capital Addis Ababa. The study area landscape is undulating and covered by a thick savannah and forest (Figure S1). Due to a lack of communications infrastructure and transportation, the possibilities to travel within the Benishangul-Gumuz region are often scarce [60]. This results in inadequate access to food and health services [61]. Also, inadequate access to safe drinking water (particularly for microbiological quality), poor sanitation, low educational level, widespread poverty, and the highest risk of under-five mortality characterize the Benishangul Gumuz region [62-67]…

The possible exposure sources to the toxic elements analyzed may be several, such as contaminated food, water, and soil pollution, especially by the abuse or misuse of chemicals in agriculture, indoor and outdoor air pollution from inefficient solid fuel combustion (wood, animal dung, charcoal, crop wastes, and coal) and burning of waste, residues of manufacturing industrial products, mining, and oil refining [81-83]. In particular, As, Cd, Hg, and Pb are known to be of health concern in Africa, including Ethiopia [81]… Water, food, air, and soil can be possible exposure sources to As from both natural geochemical processes (erosion) or anthropogenic pollution (arsenical insecticides, improper waste, and sewage disposal, combustion of fossil fuels) [81]… Human exposure to Cd in Africa is thought to be mainly due to tobacco smoking and the consumption of contaminated vegetables and crops [81]. High levels of Cd contamination were reported in lettuce grown by irrigation using water from Akaki River in Ethiopia [115]… Exposure to Hg can occur mainly by the ingestion of contaminated fish (organic Hg, such as methylmercury) and small-scale artisanal gold mining (elemental Hg) [81]. Furthermore, the gold extraction is routinely carried out near water sources, contaminating the environment, and drinking water… The reduction or removal of Pb from gasoline has produced a significant decline in pediatric morbidity. However, children and adults continue to be exposed to Pb in Africa [81]. Here, most of the Pb in the environment comes from human activities such as waste combustion, mining, indiscriminate dumping, and even the use of Pb-based paints [81]”.

These topics were added and further described in the "Results and Discussion" section.

R3: 2. The authors must inform about the water samples collected in the region. How was the collection, what was the volume collected, how was the transportation of these samples? Describe how the water was sampled (e.g., what were the methods and apparatus applied?), how were samples transported and preserved (i.e., acidified? Frozen?)

A: Thank you for your comment. The text on page 5, lines 134-142, was modified as follows: “Drinking water sampling and analyses were carried out following the Italian reference analytical methods [74,75]. Briefly, a volume of 250 mL (three replicates) of both treated and untreated water of the Blue Nile river was collected using a sterile polyethylene container with a screw cap. Each sample had an indelible identification label to identify it uniquely. Sampling was done where households got their drinking water. In particular, if there were more possibilities of access to the same water source (different access points to the Blue Nile river or different taps of the same treated water), a single sampling point was considered (Figures S2 and S3). After sampling, all water samples were transported to the laboratory using thermal bags and then filtered, acidified to 2% (v/v) HNO3 (pH <2), and stored at a temperature in the range 1-10 °C up to instrumental analysis. All water analyses were completed within one week after fieldwork. The water samples were analyzed diluted in a 1:10 and 1:100 ratio with 2% (v/v) HNO3 and without dilution. Blanks and calibration standards were also made in 2% (v/v) HNO3.”

R3: 3. The authors must cite the type of water sample that was collected and the number of samples.

A: Ok, done, thank you.

R3: 4. The abstract must be rewritten to represent the study results accordingly.

A: Thank you for your suggestion. The abstract was also modified according to the suggestions of Reviewer 4.

R3: 5. What care is taken in collecting hair samples to avoid external contamination by elements?

A: To avoid external contamination, hair samples were collected using stainless steel scissors, as reported on page 3, line 98, and disposable vinyl gloves, as previously described (Astolfi, M. L.; Protano, C.; Marconi, E.; Massimi, L.; Brunori, M.; Piamonti, D.; Migliara G.; Vitali, M.; Canepari, S. A new rapid treatment of human hair for elemental determination by inductively coupled mass spectrometry. Anal. Methods 2020, 12, 1906–1918. https://doi.org/10.1039/C9AY01871A; Astolfi, M.L.; Protano, C.; Marconi, E.; Massimi, L.; Piamonti, D.; Brunori, M.; Vitali, M.; Canepari, S. Biomonitoring of Mercury in Hair among a Group of Eritreans (Africa). Int. J. Environ. Res. Public Health 2020, 17, 1911. https://doi.org/10.3390/ijerph17061911). The sentence on page 3, lines 98-99, was modified as follows: “To avoid external contamination, hair samples (~0.05 g; ~1 cm long) were collected from the nape using stainless steel scissors and disposable vinyl gloves, as previously described [71,72]. Then, hair samples were stored in polyethylene bags until analysis at room temperature”.

R3: 6. The authors conclude that it is essential to assess children's element levels due to possible neurotoxicity involvement. In this context, the authors could be neuropsychological assessments with the study group.

A: Thank you for your comment. The toxic heavy metals (such as As, Cd, Hg, and Pb) were studied extensively, and their effects on human health are reviewed by the World Health Organization (WHO. Chemicals of public health concern in the African Region and their management: Regional Assessment Report, World Health Organization. Regional Office for Africa, Republic of Congo, 2014. Available online: https://www.afro.who.int/sites/default/files/2017-06/9789290232810.pdf. (Accessed on 31 October 2020)). It is known that exposure to toxic heavy metals even at low concentrations is associated with various health effects, including, but not limited to, neurotoxicity and carcinogenicity (ATSDR. Agency for Toxic Substances and Disease Registry. Toxicological profile for arsenic. U.S. Department of Health and Human Services, Public Health Service, Atlanta, GA. 2015; ATSDR. Agency for Toxic Substances and Disease Registry. Toxicological profile for mercury & addendum. U.S. Department of Health and Human Services, Public Health Service, Atlanta, GA. 1999, 2013; ATSDR. Agency for Toxic Substances and Disease Registry. Toxicological profile for lead. U.S. Department of Health and Human Services, Public Health Service, Atlanta, GA. 2007; ATSDR. Agency for Toxic Substances and Disease Registry. Toxicological profile for cadmium. U.S. Department of Health and Human Services, Public Health Service, Atlanta, GA. 2012; Tokar, E.J.; Benbrahim-Tallaa, L., Waalkes, M.P. Metal ions in human cancer development. Met. Ions Life Sci. 2011; 8, 375-401. https://doi.org/10.1039/9781849732116-00375). In our study, unfortunately, no information was collected that could be used for a neuropsychological evaluation. This may be an interesting topic for future research.

Reviewer 4 Report

Revision IJERPH-970059

General comments

The Manuscript entitled “Element levels and predictors of exposure in the hair of Ethiopian children”, is focused on provide updated information about the concentration of more than 40 elements in hair of children in Ethiopia. Experimental methods include inductively coupled plasma mass spectroscopy and cold vapor atomic fluorescence spectroscopy. The manuscript presents a short and limited study about the concentration of a wide group of elements in human hair. Results and its statistical analysis provide important information of general scientific interest. In general, order and structure of abstract, introduction and results sections need to be improved. Particularly Information presented in the abstract need to be improved to better describe the main content and highlights of the manuscript. Figures and tables resolution and order need to be improved for a better visualization and understanding. English language needs to be revised in all the manuscript. Details are described below

Minor revisions

Lines 21 to 24: Authors shown a range of concentrations between “essential and toxic elements”, from this point of view it is difficult to identify which elements are present in higher concentrations and which ones are in low concentrations. I understand there is too much information to be contracted but I suggest reduce it to the 3 or 4 toxic elements presents in higher concentrations and 3 or 4 essential elements present in lower concentrations.

Lines 20 to 31: It seems that there is a lot of “keywords” and in my opinion not all useful. I suggest eliminate inductively coupled plasma mass spectroscopy, cold vapor atomic fluorescence spectroscopy, non-invasive biological matrix, environmental exposure, mercury and fish consumption, keep: biomonitoring and use: human hair, children, trace elements, among others more specific than those of were proposed at the beginning.

Line 34: All elements are coming from a “natural” origin, I suggest to use the term natural or anthropogenic source.

Line 34: Please provide also examples for anthropogenic sources.

Lines 42 to 44: I think a better example of dangerous elements can be As that in fact is classified as carcinogenic to humans.

Line 46: I think there is more factor than can make children vulnerable to toxic effects such as childish activities and clarify that vulnerability can be for both environmental exposure and toxic effects.

In a quick revision I detect some studies for elements such as Hg in children hair in Ethiopia, for example https://link.springer.com/article/10.1007%2Fs12011-016-0745-9.

Lines 82 to 83: Please transform this sentence in a formula for better visualization and understanding.

Line 91: Did de authors try to say, “at least with the presence of one of the parents”.

Lines 109 to 110 and 113 to 114: Please join these two sentences regarding SM.

Line 125 (Table 1): I suggest reordering the table or add more lines to separate population characteristics; in the present from it looks like a list and difficult to understand. In fact, authors can present one table for population characteristics and another one for food consumption.

Line 126: Did the authors try to say, “(minimum and maximum)”

Lines 165 to 167: This information is repeated in lines 143 to 144.

Line 175: Did the authors try to say, “Element concentrations”

Line176 (Table 2): Please add the meaning of the acronyms at the end of the table.

Please eliminate zeros that are meaningless, for example in 0.20, can be 0.2 or 0.10 can be 0.1

Please consider using scientific notation for columns such as DL.

Please consider separate with “commas” in numbers such as 1500, can be 1,500.

Line 177: These studies were performed in population with similar characteristics, please specify.

Line 188: Please eliminate meaningless digits (1.0, can be just 1)

Figure 2: Authors should improve quality and resolution of figure 2.

Please consider use a break in X axis for a better visualization of small concentrations in upper right graph for example.

Lines 194 to 201and further sections: Here are so much information for a figure caption, please find a way to reduce or use part of the information in the graphs.

Lines 215 to 256: Here is to much important information that should be used in the abstract.

Line 257 (Table 3): Please eliminate zeros that are meaningless.

Figure 3: Please increase resolution of graphs, please consider increasing the font size.

Line 290 (Table 4): Please eliminate zeros that are meaningless.

Line 300 (Table 5): Please eliminate zeros that are meaningless.

Line 308 (Table 6): Please eliminate zeros that are meaningless.

Please add meaning of acronyms at the end of the table.

Major revisions

Lines 16 to 28: Abstract need to be improved there is some missing information regarding results.

Lines 34 a 70: Introduction need to be improved, information need to be merged in sentences and eliminate excessive new paragraphs and be ordered in a logic sequence for example:

“Elements that are present in the environment can be from both natural (e.g. volcanic activity and forest files) and anthropogenic (e.g. industry emissions, burning of fuels) [1-3]. Factors that influence human exposure and consequently the presence of possible toxic effects are mainly dietary habits (i.e. fish, meat, cereals, vegetables and water consumption), and outdoor and indoor air quality “.

Author Response

Reviewer 4 (R4)

Comments and Suggestions for Authors

Revision IJERPH-970059

General comments

The Manuscript entitled “Element levels and predictors of exposure in the hair of Ethiopian children”, is focused on provide updated information about the concentration of more than 40 elements in hair of children in Ethiopia. Experimental methods include inductively coupled plasma mass spectroscopy and cold vapor atomic fluorescence spectroscopy. The manuscript presents a short and limited study about the concentration of a wide group of elements in human hair. Results and its statistical analysis provide important information of general scientific interest. In general, order and structure of abstract, introduction and results sections need to be improved. Particularly Information presented in the abstract need to be improved to better describe the main content and highlights of the manuscript. Figures and tables resolution and order need to be improved for a better visualization and understanding. English language needs to be revised in all the manuscript. Details are described below

Minor revisions

Lines 21 to 24: Authors shown a range of concentrations between “essential and toxic elements”, from this point of view it is difficult to identify which elements are present in higher concentrations and which ones are in low concentrations. I understand there is too much information to be contracted but I suggest reduce it to the 3 or 4 toxic elements presents in higher concentrations and 3 or 4 essential elements present in lower concentrations.

Authors (A): Ok, done. Thank you for your kind review report.

R4: Lines 20 to 31: It seems that there is a lot of “keywords” and in my opinion not all useful. I suggest eliminate inductively coupled plasma mass spectroscopy, cold vapor atomic fluorescence spectroscopy, non-invasive biological matrix, environmental exposure, mercury and fish consumption, keep: biomonitoring and use: human hair, children, trace elements, among others more specific than those of were proposed at the beginning.

A: Done, thank you.

R4: Line 34: All elements are coming from a “natural” origin, I suggest to use the term natural or anthropogenic source.

A: Done.

R4: A: Line 34: Please provide also examples for anthropogenic sources.

A: Done.

R4: Lines 42 to 44: I think a better example of dangerous elements can be As that in fact is classified as carcinogenic to humans.

A: Thank you very much for your comment. We added the following sentence in the revised text: “As and Cd are classified as carcinogens (Group I), and Pb is ranked as probable carcinogens (Group IIA) in humans by the International Agency for Research in Cancer (IARC) “.

R4: Line 46: I think there is more factor than can make children vulnerable to toxic effects such as childish activities and clarify that vulnerability can be for both environmental exposure and toxic effects.

A: Again thank you for your comment. The sentence was modified as follows: “Children exposed to environmental contaminants are more susceptible than adults mainly due to their immature organs and differences in exposure [20,21]. Nutritional aspects (children eat more food and drink more water per unit of body weight than adults), rapid growth, active time spent outdoor, and specific behaviors (such as the tendency to put everything in their mouth and contact with the ground) increase the risk of exposure in children [22]. In this regard, it is important to underline that exposure to low doses of trace metals, can cause neurobehavioral and cognitive changes in children even below concentrations considered safe for most people [23]”

R4: In a quick revision I detect some studies for elements such as Hg in children hair in Ethiopia, for example https://link.springer.com/article/10.1007%2Fs12011-016-0745-9.

A: Thank you for the reference. However, the study is not related to the pediatric/adolescent age population alone and reports the Hg total data only in the hair of the following population groups: 1) students and teachers (females = 7 and males = 17) with the age range 17–49; 2) anglers and their families (females = 8 and males = 15) age range between 9 and 45, and participants with other occupation (males = 5) with the age range of 18–25. We, therefore, preferred not to add the reference to the article.

R4: Lines 82 to 83: Please transform this sentence in a formula for better visualization and understanding.

A: Done, thank you for the suggestion.

R4: Line 91: Did de authors try to say, “at least with the presence of one of the parents”.

A: Done, thank you.

R4: Lines 109 to 110 and 113 to 114: Please join these two sentences regarding SM.

A: Done, thank you for your suggestion.

R4: Line 125 (Table 1): I suggest reordering the table or add more lines to separate population characteristics; in the present from it looks like a list and difficult to understand. In fact, authors can present one table for population characteristics and another one for food consumption.

A: Done, again thank you for the suggestion. Table 1 was also modified according to the suggestions of Reviewer 1.

R4: Line 126: Did the authors try to say, “(minimum and maximum)”

A: It was a typo, thank you.

R4: Lines 165 to 167: This information is repeated in lines 143 to 144.

A: Thank you for your correction. We removed the following sentence: “GM, range and percentiles are shown for each element because the element concentration data did not fit a normal distribution”.

R4: Line 175: Did the authors try to say, “Element concentrations”

A: Done. Thank you.

R4: Line176 (Table 2): Please add the meaning of the acronyms at the end of the table.

A: Done.

R4: Please eliminate zeros that are meaningless, for example in 0.20, can be 0.2 or 0.10 can be 0.1

A: We considered a maximum of two significant digits for the standard deviation and three for the other data, reporting all results accordingly. Please note that the terminal zeros after the comma are significant: 0.400 has three significant digits and 4.0 has two significant digits.  

R4: Please consider using scientific notation for columns such as DL.

A: Done.

R4: Please consider separate with “commas” in numbers such as 1500, can be 1,500.

A: Thank you for the suggestion. We preferred not to add the comma because it increased the column width.

R4: Line 177: These studies were performed in population with similar characteristics, please specify.

A: Sorry, but we could not find the sentence.

R4: Line 188: Please eliminate meaningless digits (1.0, can be just 1)

A: Please note that 1.0 mg kg-1 (and not 1 mg kg-1) is the threshold level proposed by the National Research Council (NRC. Toxicological Effects of Methylmercury. Publisher: National Academy Press: Washington, DC, USA, 2000. Available online: https://www.ncbi.nlm.nih.gov/books/NBK225778/pdf/Bookshelf_NBK225778.pdf (accessed on 31 October 2020).

R4: Figure 2: Authors should improve quality and resolution of figure 2.

A: Thank you for your suggestion. The graphs in Figure 2 were reported separately to increase the quality and resolution of the images.

R4: Please consider use a break in X axis for a better visualization of small concentrations in upper right graph for example.

A: Done, thank you.

R4: Lines 194 to 201and further sections: Here are so much information for a figure caption, please find a way to reduce or use part of the information in the graphs.

A: Done, thank you for the suggestion.

R4: Lines 215 to 256: Here is to much important information that should be used in the abstract.

A: Thank you very much for your suggestion.

R4: Line 257 (Table 3): Please eliminate zeros that are meaningless.

A: Please see the response above.

R4: Figure 3: Please increase resolution of graphs, please consider increasing the font size.

A: Done.

R4: Line 290 (Table 4): Please eliminate zeros that are meaningless.

A: Please see the response above.

R4: Line 300 (Table 5): Please eliminate zeros that are meaningless.

A: Please see the response above.

R4: Line 308 (Table 6): Please eliminate zeros that are meaningless.

A: Please see the response above.

R4: Please add meaning of acronyms at the end of the table.

 A: Done.

Major revisions

R4: Lines 16 to 28: Abstract need to be improved there is some missing information regarding results.

A: Thank you for your suggestion. The abstract was also modified according to the suggestion of Reviewer 3.

R4: Lines 34 a 70: Introduction need to be improved, information need to be merged in sentences and eliminate excessive new paragraphs and be ordered in a logic sequence for example:

“Elements that are present in the environment can be from both natural (e.g. volcanic activity and forest files) and anthropogenic (e.g. industry emissions, burning of fuels) [1-3]. Factors that influence human exposure and consequently the presence of possible toxic effects are mainly dietary habits (i.e. fish, meat, cereals, vegetables and water consumption), and outdoor and indoor air quality “.

A: Thank you again for the suggestion. The introduction was modified accordingly.

Round 2

Reviewer 3 Report

The authors answered all questions.

Reviewer 4 Report

The Manuscript entitled “Element levels and predictors of exposure in the hair of Ethiopian children”, is focused on provide updated information about the concentration of more than 40 elements in hair of children in Ethiopia. Experimental methods include inductively coupled plasma mass spectroscopy and cold vapor atomic fluorescence spectroscopy. The manuscript presents a short and limited study about the concentration of a wide group of elements in human hair. Results and its statistical analysis provide important information of general scientific interest. After the 1st revision, the order and structure of abstract, introduction and results sections were substantially improved. Figures and tables resolution were also improved. English language still can be improved.